# A dual-labeling probe to track functional mitochondria–lysosome interactions in live cells

Qixin Chen[1,2,3,7], Hongbao Fang[1,3,7], Xintian Shao[3], Zhiqi Tian[3], Shanshan Geng[1], Yuming Zhang[1], Huaxun Fan [4], Pan Xiang[5], Jie Zhang[6], Xiaohe Tian [5], Kai Zhang [4✉], Weijiang He [1✉], Zijian Guo[1✉] & Jiajie Diao [3✉]

Mitochondria–lysosome interactions are essential for maintaining intracellular homeostasis. Although various fluorescent probes have been developed to visualize such interactions, they remain unable to label mitochondria and lysosomes simultaneously and dynamically track their interaction. Here, we introduce a cell-permeable, biocompatible, viscosity-responsive, small organic molecular probe, Coupa, to monitor the interaction of mitochondria and lysosomes in living cells. Through a functional fluorescence conversion, Coupa can simultaneously label mitochondria with blue fluorescence and lysosomes with red fluorescence, and the correlation between the red–blue fluorescence intensity indicates the progress of mitochondria–lysosome interplay during mitophagy. Moreover, because its fluorescence is sensitive to viscosity, Coupa allowed us to precisely localize sites of mitochondria–lysosome contact and reveal increases in local viscosity on mitochondria associated with mitochondria–lysosome contact. Thus, our probe represents an attractive tool for the localization and dynamic tracking of functional mitochondria–lysosome interactions in living cells.

[1] State Key Laboratory of Coordination Chemistry, Coordination Chemistry Institute, School of Chemistry and Chemical Engineering, Nanjing University, Nanjing, China. [2] Institute of Materia Medica, Shandong First Medical University & Shandong Academy of Medical Sciences, Jinan, China. [3] Department of Cancer Biology, University of Cincinnati College of Medicine, Cincinnati, OH, USA. [4] Department of Biochemistry, University of Illinois at Urbana-Champaign, Urbana, IL, USA. [5] Anhui Province Key Laboratory of Chemistry for Inorganic/Organic Hybrid Functionalized Materials, Anhui University, Hefei, China. [6] Advanced Medical Research Institute/Translational Medicine Core Facility of Advanced Medical Research Institute, Shandong University, Jinan, China. [7] The authors contributed equally: Qixin Chen, Hongbao Fang. ✉email: kaizkaiz@illinois.edu; heweij69@nju.edu.cn; zguo@nju.edu.cn; jiajie.diao@uc.edu

Mitochondria–lysosome interactions, including mitochondria–lysosome fusion (i.e., mitophagy, a process that selectively removes redundant or damaged mitochondria)[1] and mitochondria–lysosome contact (MLC)[2], are important for maintaining cellular homeostasis in eukaryotes[3,4]. Defective mitochondria–lysosome interactions are often associated with neurodegenerative diseases[5] and cancer[6]. Mitochondria and lysosomes form unique dynamic inter-organelle membrane contact sites to mediate the inter-organelle transfer of metabolites, contributing to the pathogenesis of diseases linked to the dysfunction of both organelles[2,4]. The membrane contact site between mitochondria and lysosome is defined as the contact between two different organelles in the membrane formed at close range (~10 nm), allowing them to communicate in the dynamic process from contact to separation (60 s–13 min)[2,4]. Due to the limitation of Abbe's diffraction limit (<200 nm)[7], this event cannot be captured under epi-illumination fluorescence microscopy or confocal microscopy[8]. To date, such MLC events have only been defined by the physical distance observed by super-resolution microscopy[9] and electron microscopy[2]. However, precise MLC sites cannot be directly visualized using current biochemical and cellular approaches.

Of particular interest is the development of fluorescent probes to improve the performance of super-resolution imaging in probing mitochondria–lysosome interactions[10,11]. Numerous synthetic fluorescent probes have been developed to prolong photobleaching resistance[12], reduce phototoxicity[13], decrease background signal[11], and inhibit tumor-growth[14] for imaging the morphology of subcellular organelles. However, at least two fluorescent probes with different emission spectra—small-molecule dyes[3] or fluorescent proteins[2]-are needed to monitor mitochondria–lysosome interplay. Moreover, those probes indicate only co-localization and cannot effectively track the dynamic changes of mitochondria–lysosome functional interactions, which are so valuable to understand cellular responses to stimulations and local environmental variations.

To address this problem, we developed a small-molecule fluorescent probe of hemicyanine containing propionic acid (Coupa, Fig. 1) with specific organelle targeting ability and functional fluorescence conversion capabilities. Under normal physiological conditions, Coupa attaches to lysosomes and emits red fluorescence (i.e., Coupa-lyso fluorescence); however, when exposed to reactive sulfur species (RSS) in mitochondria, Coupa is biomodified into a new compound that emits blue fluorescence (i.e., Coupa-mito fluorescence). Using structured illumination microscopy (SIM), we showed that Coupa dyes could indicate the functional interactions between mitochondria and lysosomes during mitophagy through co-localization and correlated fluorescence change. Dual-color labeling of Coupa dyes by functional fluorescence conversion can prevent the false positioning induced by the co-localization of two traditional dyes only. Moreover, we demonstrated that the viscosity-sensitive Coupa precisely localizes sites of MLC by reporting changes in local viscosity on mitochondria. With fine cell permeability, non-toxicity, and excellent photobleaching resistance, Coupa will offer new insights into the dynamic nature of mitochondria–lysosome interactions in living cells.

## Results

### Designing Coupa to track mitochondria and lysosomes.
We developed previously a family of coumarin-derived hemicyanine fluorophores possessing two intramolecular charge transfer (ICT) emission peaks (~500 and 650 nm), which provides them the chance to label mitochondria and lysosomes, respectively[15,16]. In these compounds, coumarin system contributes to emission at ~500 nm, while hemicyanine system as whole fluorophore molecule is responsible for emission at ~650 nm. Their indolenium positive charge may target them to mitochondria if they are suitably N-substituted on indolium. These hemicyanines can react with reactive sulfur species (RSSs) such as $H_2S$, $SO_2$, and GSH, which show distinct nucleophilic attacking ability in slightly alkaline microenvironment of mitochondria ($pH_m$, 8.0), to destroy the hemicyanine skeleton[15,16], and finally leave only the coumarin system to emit blue fluorescence for mitochondria labeling. With the ability of lysosomes to engulf exogenous entities, we suppose a coumarin-derived hemicyanine compound with a specific N-substituent might be engulfed by lysosomes. In the acidic lysosome microenvironment, no obvious nucleophilic agent to destroy the hemicyanine, and lysosomes can be labeled red emission of hemicyanine. Thus, it was possible to apply such a coumarin-derived hemicyanine to locate mitochondria and lysosomes with different colors simultaneously: mitochondria at approximately 500 nm and lysosomes at approximately 650 nm. After screening a series of coumarin-derived hemicyanines[15,16], we found a N-carboxylpropanyl substituted coumarin-derived hemicyanine, named Coupa (Fig. 1a), enabling such a dual labeling ability (Fig. 1b). We synthesized the probe in a single step via a Knoevenagel condensation reaction (Supplementary Fig. 1) followed by fully structure characterization with $^1H$ NMR, $^{13}C$ NMR and HR-MS (Supplementary Fig. 2–4).

To verify the luminescent properties of Coupa, we used $Na_2S$ as an $H_2S$ donor (Fig. 1c) and $Na_2SO_3$ as an $SO_2$ donor (Fig. 1f) to mimic the mitochondrial micro-environment rich in RSSs[17,18]. Coupa was excited at 405 and 560 nm respectively to determine its fluorescent response. As expected, the hemicyanine fluorescence decreased distinctly, whereas coumarin emission showed no obvious change in the presence of either RSSs. Moreover, the probe did not respond to other biologically relevant species such as $H_2O_2$, $ClO^-$, •OH, and pH alternation (Supplementary Fig. 5). These results indicate that Coupa responds to RSS by changing the conjugated structure of hemicyanine in the mitochondrial microenvironment in vitro.

The viscosity of mitochondria increases when they are damaged[19,20], which could result from reactive oxygen species (ROS) altering mobility of intracellular biomolecules[21]. Although conventional dyes (e.g., mitochondria-tracker and lysosome-tracker probes) report the morphology of mitochondria and lysosomes[3], they are insensitive to viscosity and therefore do not track the dynamic changes of an organelle's internal environment. Recently developed viscosity probes[22] can realize the imaging of the viscosity of a single organelle by using fluorescent probe with rotatable $C_{sp^2}$–$C_{sp^2}$ bond[23–25]. In Coupa, there is also $C_{2p^2}$–$C_{sp^2}$ bond in its molecule skeleton, this may provide Coupa also the fluorescent viscosity sensing ability. Then we determined the fluorescence spectra of Coupa in mixed solvents of varying viscosity by changing mixing ratio of glycerol and methanol. Higher viscosity caused enhanced fluorescence intensity under both 405-nm (Fig. 1d) and 560-nm excitation (Fig. 1g), likely arising from the limited $C_{sp^2}$–$C_{sp^2}$ rotation induced by the enhanced medium viscosity. This result suggests that both emission peaks of Coupa were sensitive to viscosity. This mechanism was confirmed by fluorescence lifetime determination (Fig. 1e, h), which revealed that both fluorescence lifetimes for coumarin and hemicyanine emissions increased with the increase of viscosity. Moreover, these results demonstrated also that the Coupa's fluorescence intensity correlates with the local medium viscosity.

### Characterization of coupa in living HeLa cells.
To characterize the imaging properties of Coupa in living cells, we applied 10 µM

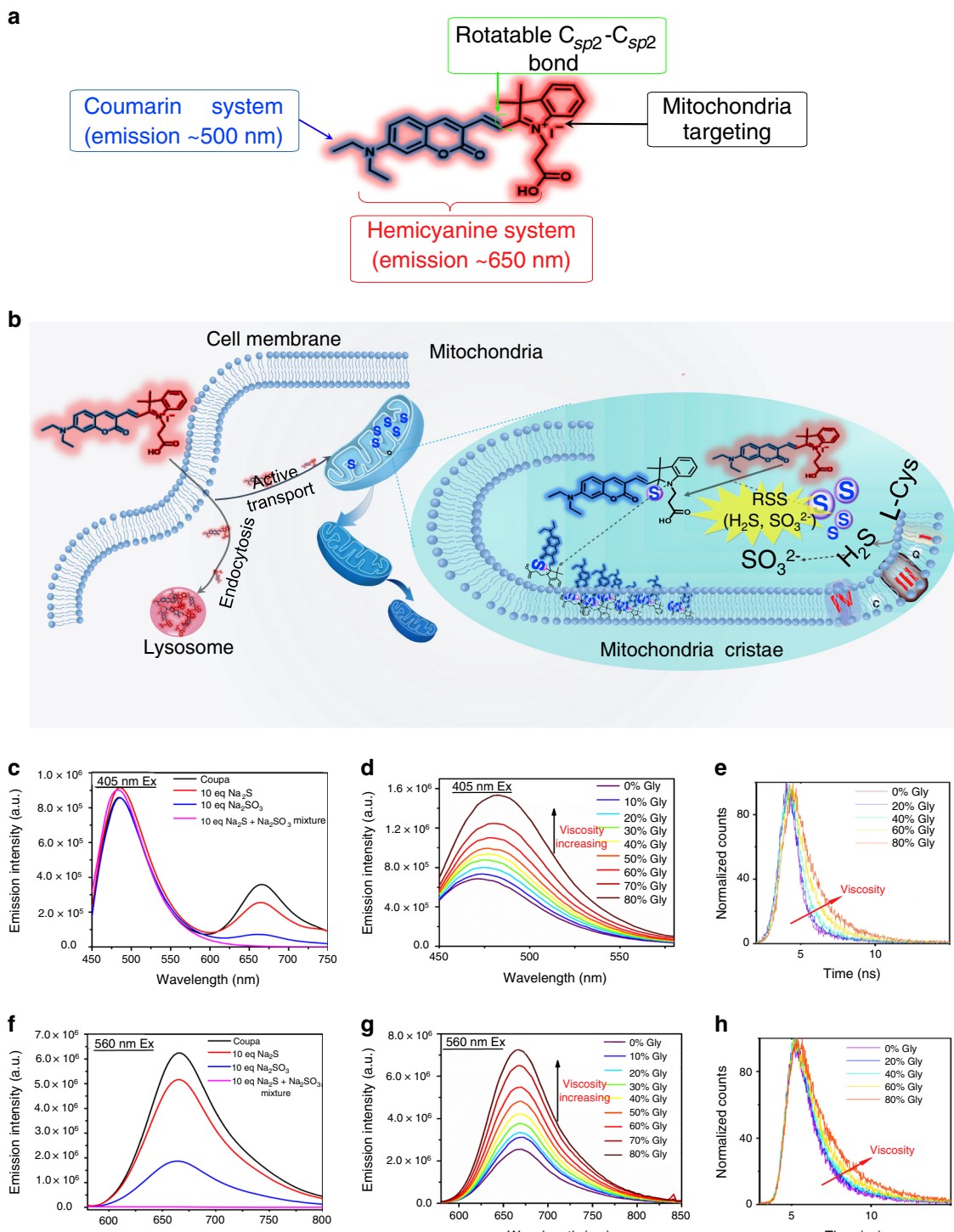

**Fig. 1 Design and fluorescence characterization of Coupa. a** Chemical structure of Coupa and the role of each functional moieties. The structure possesses two ICT emission peaks—coumarin, which peaks at approximately 500 nm, and a hemicyanine system, which peaks at approximately 650 nm—that can target mitochondria via its positive charge and respond to viscosity via altering its $C_{sp2}$–$C_{sp2}$ rotation. **b** Proposed mitochondria- and lysosome-staining mechanisms of Coupa. Coupa stains mitochondria by direct absorption, whereas lysosomal staining could result from active transport and endocytosis. Fluorescence spectra of Coupa were determined upon excitation at 405 nm (**c** and **d**) and 560 nm (**f** and **g**). **c**, **f** Fluorescence spectra determined with $Na_2S$ and $Na_2SO_3$ treatments, and $Na_2S$ and $Na_2SO_3$ completely quenched the red fluorescence (660 nm emission), leading to different Coupa emission with and without RSS. **d**, **g** Fluorescence spectra determined in media (glycerol/ methanol mixed solvent) with different viscosity; and **e**, **h** fluorescence decay profiles of Coupa in glycerol/methanol mixtures determined with picosecond pulsed excitation at 340 nm (**e**, Em at 535 nm) and 535 nm (**h**, Em at 670 nm).

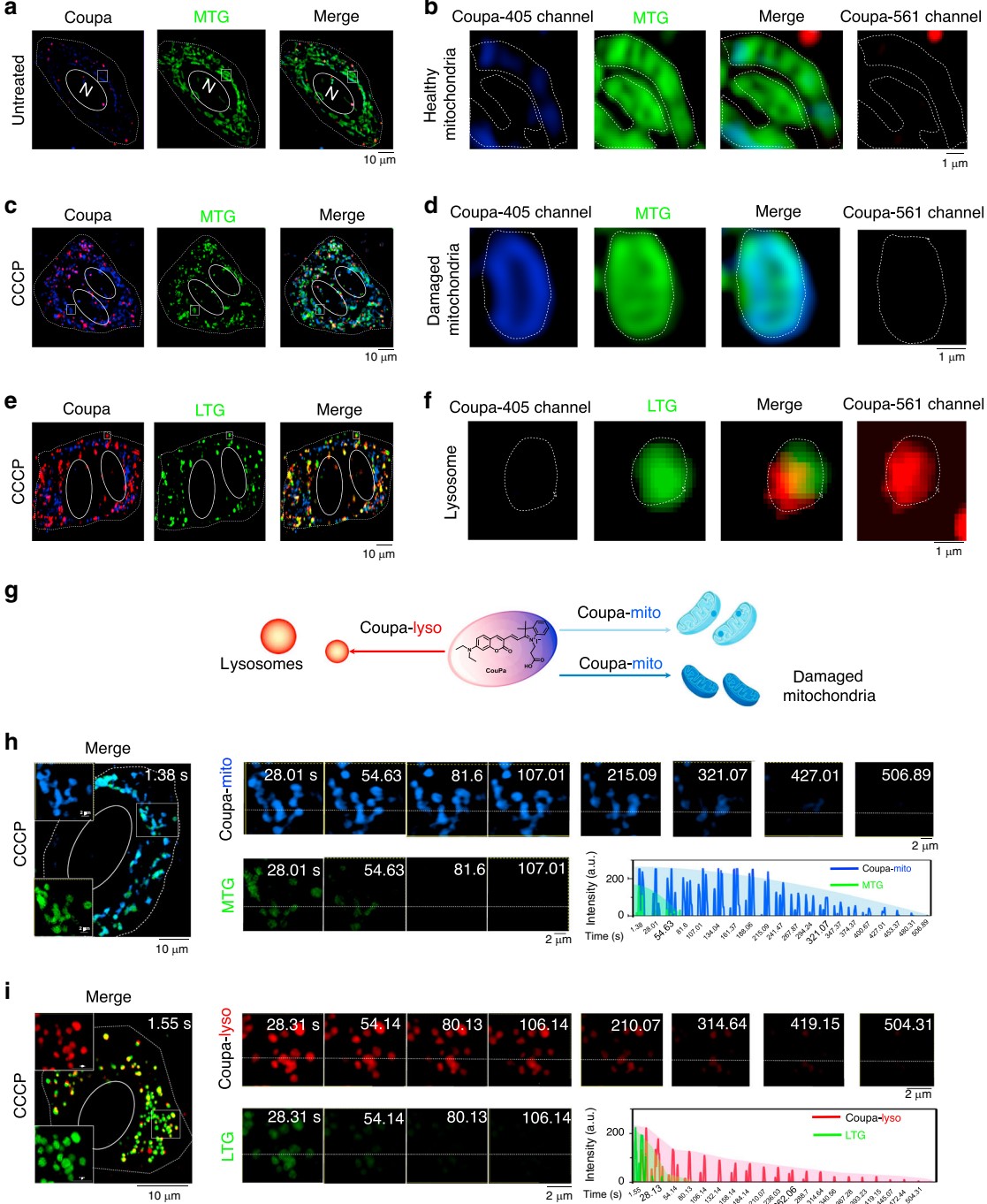

**Fig. 2 Characterization of Coupa in living cells via SIM imaging. a**, **c** Mitochondria co-stained with Coupa (405 nm excitation) and MTG (488 nm excitation) in untreated (**a**) and CCCP-treated HeLa cells (**c**). **b**; **d** Zoom-in images of white rectangles in **a** and **c**; **e** CCCP-treated HeLa cells co-stained with Coupa (561 nm excitation) and LTG (488 nm excitation); **f** Zoom-in images of white rectangle in **e**. **g** Schematic representation of Coupa staining mitochondria (Coupa-mito) and lysosomes (Coupa-lyso) simultaneously in living cells; photobleaching of Coupa-mito and MTG (**h**) and Coupa-lyso and LTG (**i**) upon continuous irradiation by 561-nm and 488-nm lasers. White dotted lines indicate region of interest for fluorescence measurement. The time-dependent fluorescence intensity was shown in the lower right panel. MTG channel: Ex, 488 nm, Em, 500–550 nm; LTG channel: Ex, 488 nm, Em, 500–550 nm; Coupa-lyso channel: Ex, 561 nm, Em, 570–640 nm, and Coupa-mito channel: Ex, 405 nm, Em, 420–495 nm.

of Coupa to stain HeLa cells for 30 min before SIM imaging using a dual channel mode with excitation at 405 nm (Coupa-405 channel, band path 420–495 nm) and 561 nm (Coupa-561 channel, band path 570–640 nm). As expected, blue particles or fibers with weak fluorescence appeared in cells in the image from the Coupa-405 channel, while a few red particles appeared in cells

in the images from the Coupa-561 channel (Fig. 2). To verify the mitochondrial targeting ability of Coupa[26], we co-stained the cells with mitochondria-tracker green (MTG) for 30 min, and the followed SIM imaging revealed that blue fluorescent particles were localized in the MTG-labeled mitochondria, whereas red particles showed a much less co-localization with mitochondria

(Fig. 2a, b, dotted white line to show the mitochondrial structure, Supplementary Fig. 6). It is clear that this Coupa-405 channel is suitable for mitochondrial tracking in living cells.

We then sought to determine the capacity of Coupa to differentiate damaged and normal mitochondria based on local viscosity changes via Coupa-405 channel imaging, since it was reported that damaged mitochondria display normally the enhanced local viscosity[27], and we confirmed this effect in HeLa cells using a previously reported mitochondria viscosity probe[20] (Supplementary Fig. 7). To damage mitochondria, we treated HeLa cells for 12 h with 10 μM of carbonyl cyanide m-chlorophenyl hydrazone (CCCP), a common mitophagy inducer[8]. Next, we stained the cells with both Coupa and MTG for colocalization imaging. The imaging revealed the filamentous mitochondria in untreated HeLa cells (Fig. 2a, b) became spherical in the CCCP-treated cells (Fig. 2c, d), consistent with the previous observations[3]. Moreover, the blue fluorescence in the damaged mitochondria was significantly higher than that in untreated HeLa cells (Fig. 2a–d and Supplementary Fig. 8), and this blue fluorescence enhancement appeared mainly on mitochondria membrane. This enhancement might origin from the mitochondria shrinkage to increase local viscosity in mitochondrial membrane. To exclude the effect of mitochondria membrane potential (MMP), we co-stained the mitochondria with Coupa and a MMP-independent mitochondria probe, mitochondria-GFP[25], and imaging with or without CCCP-altering MMP demonstrated that Coupa labeling mitochondria is not dependent on MMP, similar to that of mitochondria-GFP (Supplementary Fig. 9), and CCCP-treatment made the blue fluorescence increase with that of the MMP-independent probe GFP (Supplementary Fig. 10). Thus, Coupa-mito channel imaging with Coupa can distinguish damaged mitochondria from normal ones based on the increase of the local viscosity in mitochondria.

Red fluorescent particles displayed by Coupa-561 channel did not co-localize with MTG-labeled mitochondria with (Fig. 2d) or without (Fig. 2b) CCCP treatment. To verify that Coupa-stained red puncta (Fig. 2a, b) labeled lysosomes (Fig. 1b), we co-stained cells with Coupa and lysosome-tracker green (LTG) in CCCP-treated (Fig. 2e) and untreated cells (Supplementary Fig. 11). As expected, Coupa-labeled red particles from Coupa-561 channel image co-localized with LTG-stained lysosomes (Fig. 2f and Supplementary Fig. 11), which confirmed that Coupa could be used to label lysosomes. The average size of red spots were approximately 0.6 μm (Supplementary Fig. 12a), consistent with the reported size of lysosomes[8]. Therefore, this Coupa-561 channel is suitable for lysosomes tracking in living cells. In addition, CCCP treatment (Supplementary Fig. 12b) increased the number of red spots compared to untreated cells (Fig. 1a), consistent with the idea that the number of lysosomes increases to maintain intracellular homeostasis during mitophagy and autophagy[3]. Besides, the lysosome-labeling by Coupa was clarified by Coupa-561 channel imaging at different temperatures, metabolic, and endocytosis levels, implying that the probe entered the cells via energy-dependent endocytosis and, as such, passively targeted lysosomes (Supplementary Fig. 13). These results suggest that Coupa-labeled red particles can reflect lysosome biogenesis in physiological and pathological conditions.

All the above imaging results indicate that the conjugated system in the fluorescent probe skeleton of Coupa is immediately destroyed by RSS in mitochondria, thereby leaving the coumarin system (blue color) as the only fluorophore. Thus, through specific targeting and functional fluorescence conversion, Coupa simultaneously stains mitochondria (Coupa-mito) and lysosomes (Coupa-lyso) with blue and red fluorescence, respectively (Fig. 2g).

To characterize the photobleaching resistance of Coupa-mito, we co-stained CCCP-treated cells with Coupa and MTG and exposed them to continuous SIM laser illumination (Fig. 2h). Coupa-mito, with a photobleaching lifetime of more than 300 s, was significantly more photostable than MTG. Similarly, with a photobleaching lifetime of more than 250 s, Coupa-lyso was more photostable than LTG (Fig. 2i).

**Coupa monitors mitochondria–lysosome interactions in mitophagy.** In mitophagy, lysosomes are proposed to degrade damaged mitochondria by fusing with autophagosomes to form autolysosomes[1,28]. Conventional probes cannot track the functional interactions between the mitochondria and lysosomes undergoing mitophagy but demonstrate only the morphology of the two organelles[11,29]. To probe the dynamics of mitochondria–lysosome interactions, we co-stained the CCCP-treated cells with autophagosome dye (diacetylphloroglucinol, DAPG) and Coupa followed by SIM imaging in a triple channel mode (Coupa-405, Coupa-561 and DAPG channels). The results revealed that the blue colored mitochondria became granular after CCCP treatment (Fig. 3a, Coupa-mito) and co-localized well with the DAPG-stained autophagosomes (Fig. 3a, merge, upper right). Moreover, Coupa-labeled mitochondria showed brighter blue fluorescence when they were inside autophagosomes (Fig. 3b, 1 vs. Figure 3b, 2). By contrast, when Coupa-lyso-labeled lysosomes contacted or fused with autophagosomes (Fig. 3c, 3), the red-colored lysosomes were less fluorescent than free ones (Fig. 3c, 4). Given that Coupa-lyso and DAPG have similar photobleaching resistance (Supplementary Fig. 14), the Coupa-lyso fluorescence reduction in autolysosomes is unlikely a result of photobleaching, and Coupa reacting with RSS from the fused mitochondria might be the origin. Different from the fine co-localization between lysosome-tracker red (LTR, showing similar fluorescence intensity in both the lysosomes and autolysosomes) and DALG (an autolysosome dye) in the CCCP-treated HeLa cells (Supplementary Fig. 15), Coupa-lyso showed a low level of co-localization between lysosomes and autolysosomes. This indicates that Coupa-lyso can differentiate autolysosomes from lysosomes by showing a weakened fluorescence in the degradation steps of mitophagy and autophagy.

This different fluorescent response of Coupa-mito and Coupa-lyso might favor tracking of mitochondria-lysosomes interaction in mitophagy. Therefore, we pre-stained HeLa cells with Coupa and incubated them in color-free Dulbecco's modified Eagle's medium containing 50 μM of CCCP. The temporal SIM tracking in a dual channel mode showed a 30 min dynamic process of mitophagy from mitochondria-lysosome contact to their fusion (Fig. 3d). During the process, the red fluorescence of lysosomes gradually decreased, whereas the blue fluorescence of mitochondria increased (Fig. 3e). This result implies that Coupa is able to visualize the dynamic process of mitophagy showing the fusion-dependent, anti-correlated fluorescence changes of Coupa-mito and Coupa-lyso.

The anti-correlated fluorescence changes in the mitochondria and lysosomes undergoing mitophagy has also been observed in the Coupa-stained HeLa cells after 12 h of CCCP incubation (Fig. 3f). With the advantage of high fluorescence resolution of SIM, the lysosomes and mitochondria can be easily distinguished into three categories: lysosomes outside autolysosomes, mitochondria and lysosomes inside autolysosomes, and mitochondria outside autolysosomes. For the cells undergoing mitophagy, the SIM imaging revealed that the average fluorescence intensity of Coupa-lyso outside autolysosomes was higher than that inside autolysosomes following fusion, whereas the average intensity of Coupa-mito outside autolysosomes was less than that inside the

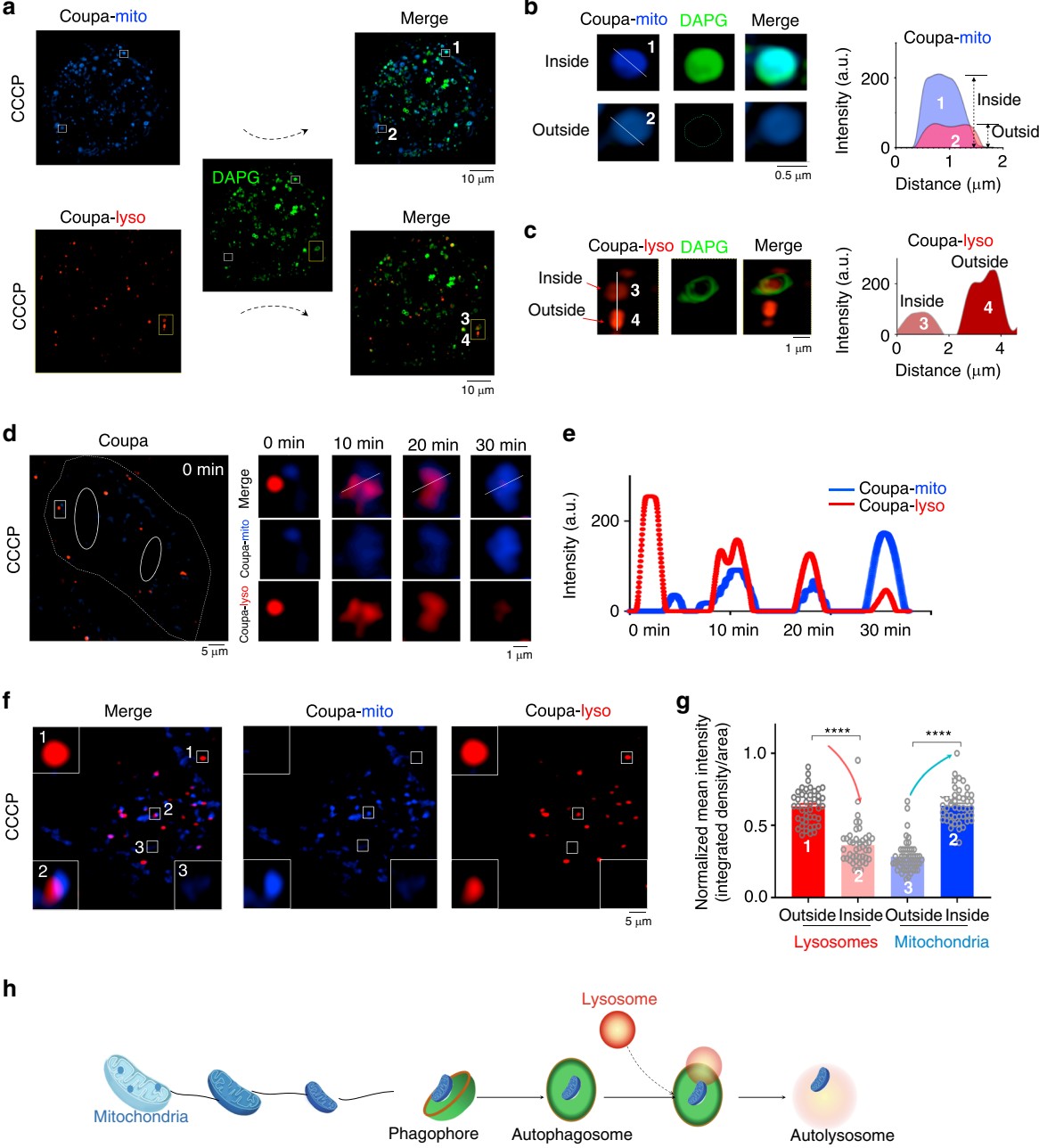

**Fig. 3 Coupa monitors mitochondrion–lysosome interactions in mitophagy via SIM imaging. a** Co-localization of Coupa-stained mitochondria/lysosomes and DAPG-stained autophagosomes in CCCP-treated (10 µM, 12 h at 37 °C) HeLa cells. **b** Zoom-in images of regions of interest in white frames representing the mitochondria inside (1) and outside (2) the autophagosomes. **c** Zoom-in images of regions of interest in white frames representing the lysosomes inside (3) and outside (4) the autophagosomes. **d** SIM tracking of mitochondrion-lysosome interaction during mitophagy in the CCCP-treated (50 µM) HeLa cells via Coupa staining; solid white lines in the merged images indicate where fluorescence intensity profiles shown in **e** were measured. **f** SIM imaging of mitochondrion–lysosome interaction in CCCP-treated cells for 12 h; white rectangles indicate lysosomes outside autolysosomes (1), mitochondria and lysosomes inside autolysosomes (2), and mitochondria outside autolysosomes (3). **g** Normalized average fluorescence intensity of mitochondrion–lysosome interactions. Data are mean ± SEM (*n* = 44 particles from 10 cells for lysosomes group, and *n* = 49 particles from 15 cells for mitochondria group). Statistical differences between the two groups were examined by Mann–Whitney test. *P* < 0.05 is considered significant (*$P$ < 0.05, **$P$ < 0.01, ***$P$ < 0.001, ****$P$ < 0.0001). Analyzed cells were obtained from three replicates. **h** Schematic representation of Coupa for monitoring mitochondria–lysosome interaction in mitophagy. DAPG channel: Ex, 488 nm, Em, 500–550 nm; Coupa-lyso channel: Ex, 561 nm, Em, 570–640 nm; Coupa-mito channel: Ex, 405 nm, Em, 420–495 nm. Source data are provided as a Source data file.

autolysosomes (Fig. 3g). These findings confirmed also the anti-correlated fluorescence changes of Coupa-mito and Coupa-lyso, suggesting that Coupa's dynamic fluorescence changes could be used to monitor mitochondria–lysosome interplay in mitophagy via a dual channel mode SIM imaging (Fig. 3h).

**Coupa reports unevenly distributed viscosity within mitochondria and identifies mitochondria–lysosome contact (MLC) sites.** The viscosity-sensitive feature of Coupa-mito fluorescence (Fig. 1d, e, Supplementary Fig. 5) prompted us to further investigate Coupa-mito's ability to monitor the viscosity in individual

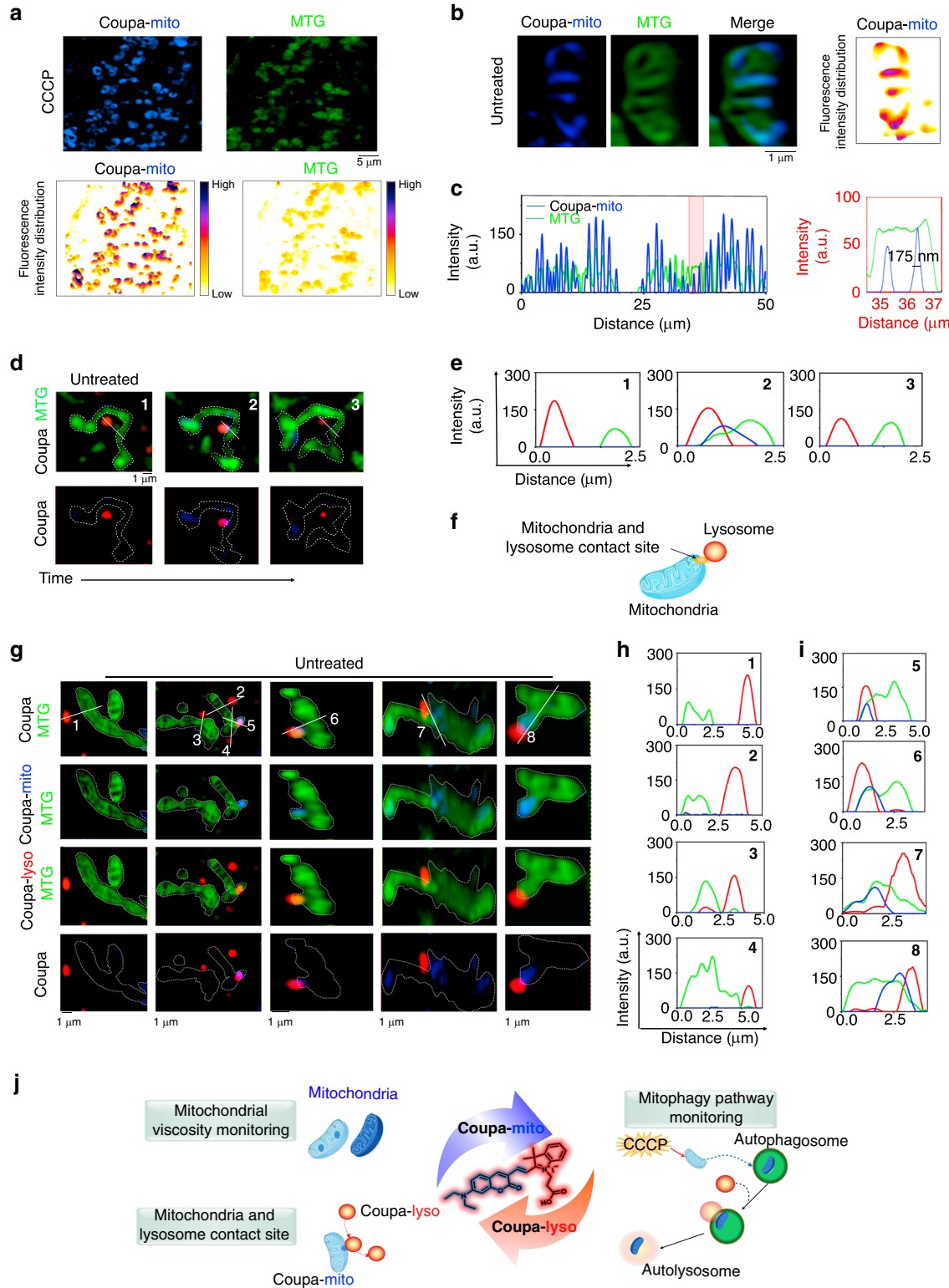

mitochondria. To avoid the influence of probe concentration, we also performed fluorescence lifetime imaging microscopy (FLIM) measurements in HeLa cells to confirm that Coupa-mito could report viscosity in live cells (Supplementary Fig. 16), and CCCP treatment significantly enhanced Coupa-mito's fluorescence lifetime corresponding to the increase of mitochondrial viscosity. We then measured the fluorescence intensity distribution of Coupa-mito to reveal the viscosity profile inside mitochondria, and the

fluorescence intensity of Coupa-mito staining mitochondria in CCCP-treated cells was accordingly much higher than that of MTG-stained mitochondria (Fig. 4a, Supplementary Fig. 17) confirming the local viscosity enhancement in damaged mitochondria induced by CCCP. To obtain a viscosity distribution of mitochondrial fine structure before damage induced by CCCP, we applied MTG and Coupa to co-stain mitochondria in untreated HeLa cells. Colocalization imaging via SIM revealed that the

**Fig. 4 Coupa reports unevenly distributed viscosity within mitochondria and identifies mitochondria–lysosome contact (MLC) sites via SIM imaging.**
**a** Fluorescence distribution of mitochondria stained with Coupa-mito and MTG in CCCP-treated cells; **b** the distribution of mitochondria cristae in untreated HeLa cells co-stained with Coupa-mito and MTG. **c** Distribution of fluorescence intensity on the solid lines across mitochondria (showed in Supplementary Fig. 20) stained with Coupa-mito and MTG, showing a spatial resolution of Coupa-mito-stained mitochondria up to 175 nm. **d** The dynamic process of mitochondria and lysosome contact in untreated HeLa cells stained by MTG and Coupa-mito; and **e** the related distribution of fluorescence intensity on solid white lines in **d**, showing the appearance of blue fluorescence as the lysosome approaches the mitochondria. **f** Schematic representation of MLC contact site. **g** MLC events revealed in untreated HeLa cells co-stained with Coupa and MTG. 1–4 show areas with no MLC events and solid white lines correspond to fluorescence intensity shown in **h**, while 5–8 show representative MLC events and solid white lines correspond to fluorescence intensity shown in **i**. **j** Schematic representation of the application of Coupa for synchronously monitoring mitochondria–lysosome interactions. MTG channel: Ex, 488 nm, Em, 500–550 nm; Coupa-lyso channel: Ex, 561 nm, Em, 570–640 nm; and Coupa-mito channel: Ex, 405 nm, Em, 420–495 nm.

entire mitochondrial cristae structure could be clearly displayed by MTG fluorescence with the cristae-to-cristae distance ranging from 100 to 200 nm (Fig. 4b and Supplementary Fig. 18), whereas only a few areas of the mitochondrial cristae were stained by Coupa-mito, which confirmed that Coupa targets the inner membrane of mitochondria (Figs. 1b and 4b). We also analyzed the distribution of fluorescence of Coupa-mito in mitochondrial cristae, which revealed also an uneven distribution in mitochondria of untreated cells (Fig. 4b, right).

To rule out the possibility that the uneven distribution of Coupa-mito fluorescence within the mitochondrial membrane was caused by poor fluidity of Coupa, we performed fluorescence recovery after photobleaching (FRAP) experiments in CCCP-treated HeLa cells (Supplementary Fig. 19). We found the fluorescence recovery of Coupa-mito fluorescence in mitochondria was faster than that of MTG-stained ones (MTG), which suggests that the uneven fluorescence distribution was not caused by Coupa's poor fluidity. To compare the resolution of MTG- and Coupa-stained mitochondria, we measured the fluorescence intensity along the solid yellow lines shown in Supplementary Fig. 20. Coupa-mito's fluorescent response to viscosity showed high resolution with full width at the half-maximum up to 175 nm (Fig. 4c). Our results confirm that Coupa-mito can allow the visualization of mitochondrial viscosity with a high resolution of up to 175 nm and excellent photobleaching resistance of up to 400 s.

Finally, we hypothesized that the blue fluorescence from Coupa-mito would increase as lysosomes approach mitochondria to form effective MLC sites, at which proteins cluster may result in local viscosity enhancement. To test this hypothesis, we measured the Coupa-mito fluorescence as lysosomes approached mitochondria in live cells. We co-stained HeLa cells with Coupa and MTG for dual channel SIM imaging (Fig. 4d). As expected, Coupa-mito (blue color) always located between the mitochondrion (green) and lysosome (red) (Fig. 4e) at the site of the MLC event (Fig. 4g, i 5–8). It is noted that no Coup-mito (blue color) appeared in the area without MLC (Fig. 4g, h 1–4). All these indicate that Coupa can precisely identify MLC sites.

## Discussion

Here, we developed Coupa, a dual-labeling probe enabling the simultaneous labeling of mitochondria and lysosomes in living cells through functional fluorescence conversion. Compared with other mitochondria and lysosome probes[10,11], or commercial mitochondria and lysosome dyes[3], Coupa is able to report the detailed intermediates of functional mitochondria-lysosome crosstalk because of its anti-correlated fluorescent signal change (increase of the blue fluorescence of Coupa-mito and decrease of the red fluorescence of Coupa-lyso, Fig. 3g, h) during mitophagy. We envision that Coupa will benefit both fundamental and translational research in delineating regulatory mechanisms of specific steps during mitophagy, as well as simplifying the staining process for large-scale screening.

Because of its unique property of reporting localized viscosity in mitochondria, Coupa can be used to distinguish and pattern different viscosities in mitochondria, as well as evaluate and screen potential drugs for targeting mitochondrial viscosity or protection in cells. Moreover, Coupa revealed that MLC sites were associated with increased local viscosity of mitochondria, which cannot be reported by other viscosity probes[30]. Previously, MLC was defined physically as a stable contact between the membranes of mitochondria and lysosomes in close apposition (~10 nm) without fusion[4]. Here, the local viscosity increase indicated by our Coupa dye represents a biological characteristic of MLC, which might be caused by clustering of proteins involved in MLC process. Future studies will reveal the relationship between mitochondrial proteins and MLC.

Compared with reported principles of multicolor probes (such as mt-Keima[31] and TPA-BTTDO[32]), Coupa provides a novel design strategy of using the unique internal environment, RSS in mitochondria, to convert the structure of the probe so that it emits a different color, enabling multicolor labeling. This design principle could be generalized for the development of multicolor probes that monitor interactions of mitochondria with other organelles such as endoplasmic reticulum, Golgi apparatus, and lipid droplets.

## Methods

**Synthesis and characterization of Coupa dye**. The coumarin aldehyde derivative (Compound 1) and indole derivatives (Compound 2) were prepared by following methods reported in the literature[15]. Compound 1 (245 mg, 1.0 mM) and Compound 2 (359 mg, 1.0 mM) were mixed in 10 mL $CH_3CN$. After the solution was refluxed and stirred overnight, the solvent was evaporated via reduced vacuum. The crude product Coupa was separated by gel chromatography with the eluent as $CH_2Cl_2/CH_3OH$ (50/1, $v/v$) and the dark blue product was obtained as the yield 40%. $^1H$ NMR (400 MHz, $CD_3OD$, $\delta$, ppm) $\delta$ 8.58 (s, 1H), 8.31 (d, $J = 15.7$ Hz, 1H), 8.05 (d, $J = 15.8$ Hz, 1H), 7.78 (d, $J = 8.1$ Hz, 1H), 7.71 (d, $J = 7.0$ Hz, 1H), 7.63 – 7.52 (m, 3H), 6.91 (dd, $J = 9.1$, 2.4 Hz, 1H), 6.62 (d, $J = 2.2$ Hz, 1H), 4.74 (t, $J = 7.2$ Hz, 2H), 3.60 (q, $J = 7.1$ Hz, 4H), 2.82 (t, $J = 7.2$ Hz, 2H), 1.81 (s, 6H), 1.27 (t, $J = 7.1$ Hz, 6H). $^{13}C$ NMR (151 MHz, MeOD) $\delta$ 181.93, 176.49, 160.68, 158.48, 155.09, 150.71, 143.66, 141.25, 135.65, 132.76, 130.61, 129.31, 128.99, 122.88, 114.55, 112.71, 111.67, 110.45, 106.37, 96.75, 52.07, 45.46, 44.49, 43.69, 34.50, 26.10, 11.79. HRMS (positive mode, $m/z$): calcd. 459.2278, found 459.2276 for [M-I]$^+$.

**Cell culture**. HeLa cells were cultured in Dulbecco's modified Eagle's medium (#11965118, DMEM, Thermo Fisher Scientific) supplemented with 10% fetal bovine serum (#26140079, FBS, Thermo Fisher Scientific), penicillin (100 units/ml), and streptomycin (100 µg/ml; #15140163, 10,000 units/ml, Thermo Fisher Scientific) in a 5% $CO_2$ humidified incubator at 37 °C.

**Cell treatment and staining**. A total of $2 \times 10^5$ cells were seeded on a glass-bottom micro-well dish and incubated with 2 ml of DMEM supplemented with 10% FBS for 24 h, then stained with Coupa (10 µM) for 30 min and with 100 nM Mito-Tracker Green (#M7514, MTG, Invitrogen) or 200 nM lyso-Tracker Green (# L7526, LTG, Invitrogen) at 37 °C for another 30 min, followed by 10 µM CCCP for 12 h. After treatment, the cells were washed 3 times with pre-warmed free DMEM, and washed with free DMEM 3 times. Finally, cells were cultured in phenol-free medium (#1894117, Gibco) and observed under a confocal laser scanning microscopy (LSM-710, Carl Zeiss, Inc.) or Nikon-SIM super-resolution microscope (Tokyo, Japan).

**Confocal laser scanning microscopy**. The images were obtained using an LSM-710 confocal laser scanning microscope (Carl Zeiss, Inc.) equipped with a 63×/1.49 numerical aperture oil immersion objective lens and were analyzed with ZEN software (version 2012 SP1, Carl Zeiss, Inc.) and ImageJ software (version 1.51j8, National Institutes of Health).

**SIM super-resolution microscopy imaging**. Super-resolution images were acquired on a commercial Nikon-SIM Microscope (version AR5.11.00 64 bit, Tokyo, Japan). Images were obtained at 512 × 512 using Z-stacks with a step size of 0.2 μm. All fluorescence images were analyzed and their backgrounds were subtracted with ImageJ software."

**Fluorescence recovery after photobleaching (FRAP) and photobleaching assay**. FRAP experiments were performed on an LSM-710 confocal laser scanning microscope (Carl Zeiss, Inc.) microscope with a 60× oil immersion objective. The cell was stained with MTG and Coupa for 375 s using a laser intensity of 100% at 488 nm for MTG and 405 nm for Coupa. A recovery image was obtained at a 15-s interval. The fluorescence intensity of the photobleached area was normalized to the intensity of the unbleached area.

Photobleaching experiments were performed on a commercial Nikon-SIM Microscope (Tokyo, Japan) with a 100× oil immersion objective. The cell was stained with Coupa and MTG or Coupa and LTG after it was exposed to Nikon-SIM lasers intensity of 100% 405 nm for Coupa-mito, 488 nm for MTG and LTG, or 561 nm for Coupa-lyso. The image was obtained at a 30-s interval. SIM images were analyzed with Nikon Elements and ImageJ software.

**Cellular uptake assay**. HeLa cells were incubated by 10 μM Coupa under different conditions. 37 °C: the cells were incubated with Coupa at 37 °C for 30 min. 4 °C: the cells were incubated with Coupa at 4 °C for 30 min. MI (Metabolic inhibitors): the cells were pre-incubated with 2-deoxy-D-glucose (50 mM) and oligomycin (5 μM) in FBS-free DMEM at 37 °C for 1 h and then incubated with Coupa at 37 °C for 30 min. $NH_4Cl$:the cells were pre-incubated with $NH_4Cl$ (50 mM) in FBS-free DMEM at 37 °C for 1 h and then incubated with Coupa at 37 °C for 30 min.

**Fluorescence lifetime imaging microscopy (FLIM)**. For FLIM, HeLa cells were incubated in DMEM medium containing 10% FBS and 1% antibotics (penicillin and streptomycin), and maintained at 37 °C in an atmosphere of 5% $CO_2$ and 95% air. Then, HeLa cells were cultured and stained with Coupa (10 μM) for 30 min and washed by DMEM for 3 times. 10 μM CCCP were used for 12 h. FLIM studies were carried out by using an inverted-type laser scanning confocal microscope (Leica TCS SP8 confocal microscopy) with a 100× oil immersion objective. The incubated cells were excited at 405 nm, and time-correlated single-photon counting (TCSPC) technique was used to count emission photons, and record pictures of cells in the entire field of view. The exponential reconvolution fitting for the fluorescence decays was used to obtain the fluorescence lifetime.

**Data analysis**. Statistical analysis was performed with Prism 8 (GraphPad). Normality and lognormality test was conducted. In the case of normal distribution, the statistical comparison of results was checked with a Student's *t* test. In the case of non-normal distribution, the statistical comparison of results was checked with a Mann-Whitney test. The levels of significance were set at n.s. (no significant difference), $*P < 0.05$, $**P < 0.01$, $***P < 0.001$, and $****P < 0.0001$. Data are presented as mean ± SEM. SEM was used to compare experimental results with controls[33]. For co-localization analysis and average fluorescence intensity, we used individual fluorescent spots or randomly divided cells into four sub-cellular regions for quantitative analysis. Analyzed cells were obtained from three replicates. Statistical significances and sample sizes in all graphs are indicated in the corresponding figure legends. All associated data points can be found in the Source Data file.

**Statistics and reproducibility**. Each experiment was repeated three times independently with similar results. All images shown are representative results from biological replicates

**Reporting summary**. Further information on research design is available in the Nature Research Reporting Summary linked to this article.

## Data availability

All data are available from the corresponding author on reasonable request. The original files of all images included in figures have been deposited in Zenodo with the identifier https://doi.org/10.5281/zenodo.4120402. Source data are provided with this paper.

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

## Acknowledgements

W.H. and Z.G. were supported by the National Natural Science Foundation of China (21977044 and 21731004) and the Excellent Research Program of Nanjing University (ZYJH004). K.Z. was supported by the University of Illinois at Urbana–Champaign. J.D. was supported by the Department of Cancer Biology, University of Cincinnati College of Medicine. Q.C. was supported by academic promotion program of Shandong First Medical University (2019LJ003). We thank Dr. Junling Yin at Jinan University for gifted mitochondria-viscosity probe as a control, Dr. Yanan Zhao at Shandong University for assistant on statistical analysis, Translational Medicine Core Facility of Shandong University for consultation and instrument support, Dr. Yi Cao at Nanjing University for fruitful discussions, and Life Science Editors for editing services.

## Author contributions

Q.C. and H.Fang collected all 3D-SIM super-resolution microscopy data. Q.C. and X.S. analyzed and processed the SIM data. Q.C. and Z.T. cultured cell. H.Fang, S.G., and Y.Z. synthesized and characterized Coupa. H.Fan, Y.Z., and J.Z. performed confocal laser scanning microscopy. X.T. and P.X. performed FLIM assay. K.Z., W.H., Z.G., and J.D. conceived the project, designed the experiments, and wrote the manuscript with the help of all authors.

## Competing interests

The authors declare no competing interests.
