## [Peer Review File · Nature Communications]

Reviewers' comments:

Reviewer #1 (Remarks to the Author):

The authors introduced a cell-permeable, biocompatible, viscosity-responsive, small organic molecular probe, Coupa, to monitor the interaction of mitochondria and lysosomes in living cells. Rich biological experiments were done. From the perspective of probe design and bioimaging purpose, this work is novel and will draw the interest of the community. An original conclusion was drawn on the localization and dynamic tracking of functional mitochondria-lysosome interactions in living cells, which will influence thinking in the field. Thus, I recommend this work to be published in this journal after addressing the following concern:

(1) Structured illumination microscopy (SIM) was used in this work. Compared to confocal fluorescence imaging, is there any difference? Please explain the advantage.

(2) RSS exist not only in the mitochondria but also the cytoplasm of the live cells and the probe could also distribute in the cytoplasm, so the fluorescence in the blue region most probably does not occur only in the mitochondria targetably.

(3) What is the theoretical basis to state that the viscosity of mitochondria increases when damaged? The reference to support CCCP as a mitophagy inducer should be provided.

(4) When the cells were damaged with CCCP and mitochondrial membrane potential would change, can the probe still target mitochondria? This effect should be elaborated.

(5) In line 126 "mitochondrial" should be mitochondria ?

(6) There is no definition on LTR in line 159.

(7) Some latest new fluorescence probe on viscosity should be added (such as Chem. Commun., 2019, 55, 7410-7413; J. Am. Chem. Soc. 2019, 141, 45, 18301).

Reviewer #2 (Remarks to the Author):

This manuscript by Qixin Chen and colleagues reports on Coupa, a small organic probe that can be used to image mitochondria and lysosomes in living cells.

Coupa is reported to specifically label these two organelles, thereby emitting blue and red fluorescence. Moreover, it is reported to allow to read out sub-mitochondrial viscosity.

Indeed, such a probe is missing and I was excited to review this manuscript. The manuscript is very well written.

Unfortunately, in my view, the key statements of this manuscript are not substantiated by the presented data. In the current form this manuscript is not suitable for publication.

In the following I give some of my concerns.

Almost all conclusions in this manuscript rely on the analysis of co-localization. However, for all conditions only a single image is shown. No quantifications are reported. For example, already visual inspection of the first image (Fig. 2a, presumably a well-chosen example image), which should report on the co-localization of Coupa-Mito and MTG, is in the referee's eyes not convincingly showing a high level of co-localization of the two signals.

Without a solid statistical analysis of the level of co-localization throughout the manuscript, it is impossible to follow the conclusions.

In fact, the only statistical analysis is reported in Fig. 3g. Apparently, three manually picked fluorescent dots within a single cell were chosen for statistical analysis. This is just not up to the standards of the field.

The authors state that the viscosity of mitochondria increases when they are damaged. This would certainly be interesting. Unfortunately, the paper that is cited for this statement (Lit 16) does not provide any experimental data to support this notion.

The authors report the finding that different concentrations of glycerol (0-80 %) affect the fluorescence intensity of Coupa. They further found that in apoptotic mitochondria Coupa seems to increase in brightness, leading to the conclusion that the viscosity in apoptotic mitochondria changes and Coupa is able to report on these changes.

The authors entirely ignore other, much more obvious explanations: The increase could just be a geometrical effect, as during apoptosis thin mitochondrial tubules are transformed into large spheres, which are inherently brighter. Moreover, it is well documented that binding of MTG to the mitochondrial inner membrane is depending on the membrane potential, which breaks down upon apoptosis. Hence measuring the ratio between Coupa and MTG appears not to be the best measure. It is surprising that alternative and obvious other explanations for the change of the fluorescence intensity of Coupa are not addressed.

This makes the further use of Coupa to analyze sub-mitochondrial viscosity even more questionable. Indeed, it is unclear why the EM image (Fig. 3i) shows a crista-to-crista distance of about 100 nm, whereas in the SIM-image (Fig. 4b) the crista-to-crista distance seem to be rather at 1000 nm. Also here, for valid conclusions, a sound statistical analysis would be essential.

Reviewer #3 (Remarks to the Author):

The manuscript by Chen et al describes a new fluorescent probe that targets mitochondria and lysosomes in different emission colors. The probe is based on a hemicyanine derivative, which exhibits dual emission dependent on the environment. The dual intracellular localization is claimed to help visualizing interaction of mitochondria with lysosomes during mitophagy. Although the results look interesting, the data interpretation is speculative. Generally, the authors do not provide sufficient experimental evidence for their claims. Therefore, I do not recommend this manuscript for publication in Nature Communications. My detailed arguments are given below. After some revisions it could be recommended to another journal, such as Scientific Reports.

1) The key claim of this work is that the new probe is sensitive to viscosity. Indeed, experiments in solvents suggest some sensitivity to viscosity in solvents. However, this sensitivity is limited and it does not really mean that the variation of intensity observed inside the cells are related to viscosity. In fact, the changes in the intensity could be related to multiple reasons, including inhomogeneity in dye local concentration, dye self-quenching, dye aggregation, etc. Therefore, interpretation of data in multiple places of the manuscript (abstract, then in pages 6, 7, 9-12) is speculative. One way to provide the evidence that the observed intensity changes are linked to the viscosity is the fluorescence lifetime imaging (FLIM) combined with calibration measurements of lifetimes in organic solvents. Alternatively, the author can remove/revise claims related to viscosity sensing and submit this manuscript to a different journal.

2) Another problem concerns reactive sulfur species (RSS). The authors showed that the long-wavelength band of the dye can be quenched by RSS, but this quenching is not so efficient. In the figure 1c and 1d, it can be seen that in the presence of RSS, the red emission is still stronger than the blue emission. Therefore, it is hard to understand that this partial quenching can be the main reason for the blue emission in mitochondria (as claimed in pages 5 and 7) without any contribution of the red. It seems that the reason for the blue emission of this dye in mitochondria remains unclear, which poses serious problem for interpretation of the rest of the results in the present work. This is the second key reason why this work cannot be recommended for the present journal.

3) Presentation of electron microscopy (EM) data is misleading. Indeed, Figure 3i shows mitochondria and lysosomes, but the authors cannot claim that the structures observed in EM "correlate" with emission of the probe. The authors have not performed correlative EM-fluorescence measurements, therefore mentioning this correlation is the overstatement.

4) In all figure captions presenting multi-color images, the details on the excitation and emission wavelength should be provided.

5) Page 5, Introduction: "The viscosity of mitochondria increases when they are damaged." This

statement requires references to corresponding literature.

6) Page 5, Introduction: "Although conventional dyes (e.g., mitochondria-tracker and lysosome-tracker probes) report the morphology of mitochondria and lysosomes³, they are insensitive to viscosity and therefore do not track the dynamic changes of an organelle's internal environment". Viscosity sensitive probes for organelles have already been developed (see for instance: Chambers et al ACS Nano 2018, 12, 5, 4398). Therefore, this sentence should be revised and relevant literature should be cited.

7) Page 5: "mechanism of multicolour labelling of mitochondria and lysosomes is shown in Fig. 1b". The authors have not really studied the mechanism of the labelling. Here, they only show a proposed or expected mechanism.

Response to the review comments

We thank three reviewers for their constructive criticism and also for noting the importance of our study. We have revised the manuscript accordingly and all of their concerns were addressed by appropriate revision of the text. We believe the manuscript greatly improved in its rigor and presentation thanks to the reviewers' help.

Reviewers' comments:

Reviewer #1

The authors introduced a cell-permeable, biocompatible, viscosity-responsive, small organic molecular probe, Coupa, to monitor the interaction of mitochondria and lysosomes in living cells. Rich biological experiments were done. From the perspective of probe design and bioimaging purpose, this work is novel and will draw the interest of the community. An original conclusion was drawn on the localization and dynamic tracking of functional mitochondria-lysosome interactions in living cells, which will influence thinking in the field. Thus, I recommend this work to be published in this journal after addressing the following concern:

We thank the reviewer for the positive note on our manuscript and valuable comments.

Comment 1. *Structured illumination microscopy (SIM) was used in this work. Compared to confocal fluorescence imaging, is there any difference? Please explain the advantage.*

Response: Due to the limitation of Abbe diffraction, the structure of subcellular organelles below 200 nm cannot be observed by traditional fluorescence microscope. Compared with the traditional fluorescence microscope and confocal microscope, SIM extends the research scope of cell biology from the cell level to the nanometer scale, which enables researchers to observe the dynamic nano-scale ultrastructure, which is why SIM is used as the research tool in this manuscript. We have added a description of SIM advantage in the introduction section. It reads now (line 45-49):

“The membrane contact site between mitochondria and lysosome, MLC, is defined as the contact between two different organelles in the membrane formed at close range (~10 nm), allowing them to communicate in the dynamic process from contact to separation (60 s - 13 min). **Due to the limitation of Abbe's diffraction limit (< 200 nm)⁷, this event cannot be captured under epi-illumination fluorescence microscopy or confocal microscopy⁸.**”

Comment 2. *RSS exist not only in the mitochondria but also the cytoplasm of the live cells and the probe could also distribute in the cytoplasm, so the fluorescence in the blue region most probably does not occur only in the mitochondria targetably.*

Response: Thanks for the comment. Indeed, RSS exists in other organelles besides mitochondria. In Coupa, we introduced a cyanine structure contains lipophilic positive charge (Chen et al., 2015; Li et al., 2020) for specific mitochondria targeting by electrostatic interaction (Figure 1a and 1b). On the other hand, the coumarin-derived hemicyanines react with only reagent showing distinct nucleophilic attacking ability. It is mitochondria's alkaline microenvironment of pH_m 8.0 that made RSS in mitochondrial possess high nucleophilicity suitable for reacting hemicyanine structure. While RSS in other organelles or plasma, their nucleophilicity might be not so strong enough to destroy hemicyanine. Therefore, our imaging results revealed that blue region can only be imaged in the mitochondria, and no blue area in the cytoplasm or other organelles was found (Figure 2).

Comment 3. *What is the theoretical basis to state that the viscosity of mitochondria increases when damaged?*

Response: We thank the reviewer for this comment. Viscosity is an important microenvironment-related parameter in mitochondria that plays an essential role in many biological behaviors (Ren et al., 2017; Yang et al., 2014). These variations are caused by changing the mobility of biomolecules within the cells, such as the diffusion of reactive oxygen species (ROS) during an oxidative stress attack (Sun et al., 2018). As it is generally accepted that, when mitochondria are damaged, ROS increases (Sabharwal and Schumacker, 2014) and then improves viscosity through oxidation of cellular components (Sommer et al., 2015). We added theoretical basis for the viscosity of mitochondria increases when damaged (line 109-113). It reads now:

“The viscosity of mitochondria increases when they are damaged^{19,20}, which could result from the reactive oxygen species (ROS)-mediated change of mobility of intracellular biomolecules²¹. Although conventional dyes (e.g., mitochondria-tracker and lysosome-tracker probes) report the morphology of mitochondria and lysosomes³, they are insensitive to viscosity and therefore do not track the dynamic changes of an organelle’s internal environment.”

Comment 4. The reference to support CCCP as a mitophagy inducer should be provided.

Response: We have added a citation as Ref 8 (line 141-143). It reads now:

“To damage mitochondria, we treated HeLa cells for 12 h with 10 μ M of carbonyl cyanide m-chlorophenyl hydrazone (CCCP), a common mitophagy inducer⁸.”

Comment 5. When the cells were damaged with CCCP and mitochondrial membrane potential would change, can the probe still target mitochondria? This effect should be elaborated.

Response: We thank the reviewer for this question, since traditional mitochondrial membrane potential (MMP) experiments rely on JC-1 detection, which coincides with our probe in red channel, thus we designed another experiment to clarify this issue.

A commercial mitochondrial fluorescent protein (Mitochondria-GFP, a fusion construct of the Leader sequence of E1 alpha pyruvate dehydrogenase and emGFP, independently of MMP) (Miyake et al., 2012) was selected as a control to image mitochondria, and then CCCP was applied to destroy MMP. Next the co-localization of our probe (Coupa-mito) and mitochondria-GFP was investigated with or without CCCP altering MMP (Figure S9a). Result showed that the mitochondria labeled by Coupa-mito and Mitochondria-GFP were co-located with or without CCCP altering MMP (Figure S9b), and no significant difference was found (Figure S9c). This result proved that Coupa-mito labeling mitochondria is independent of MMP.

Figure S9 in the revised Supporting Materials

We have added this result as the new Figure S9 in Supporting Materials. And the related discussion has been added in the text (line 149-154), and it reads now:

“To exclude the effect of mitochondria membrane potential (MMP), we co-stained the mitochondria with Coupa and a MMP-independent mitochondria probe, mitochondria-GFP²⁵, and imaging with or without CCCP-altering MMP demonstrated that Coupa labeling mitochondria is not dependent on MMP, similar to that of mitochondria-GFP (Fig. S9), and CCCP-treatment made the blue fluorescence increase with that of the MMP-independent probe GFP (Fig. S10).”

Comment 6. In line 126 “mitochondrial” should be mitochondria ?

Response: We have corrected this mistake accordingly (line 155).

Comment 7. There is no definition on LTR in line 159.

Response: We have defined LTG as lysosome-tracker red (LTR) (line 199).

Comment 8. Some latest new fluorescence probe on viscosity should be added (such as *Chem. Commun.*, 2019, 55, 7410-7413; *J. Am. Chem. Soc.* 2019, 141, 45, 18301).

Response: Thanks for your comment, and we have added these citations (reference 19 and 23). It reads now:

“The viscosity of mitochondria increases when they are damaged^{19,20}, which could result from the reactive oxygen species (ROS)-mediated change of mobility of intracellular biomolecules²¹. Although conventional dyes (e.g., mitochondria-tracker and lysosome-tracker probes) report the morphology of mitochondria and lysosomes, they are insensitive to viscosity and therefore do not track the dynamic changes of an organelle’s internal environment. Recently developed viscosity probes²² can realize the imaging of the viscosity of a single organelle by using fluorescent probe with rotatable Csp2–Csp2 bond^{23,24,25}.”

Reviewer #2 (Remarks to the Author):

This manuscript by Qixin Chen and colleagues reports on Coupa, a small organic probe that can be used to image mitochondria and lysosomes in living cells.

Coupa is reported to specifically label these two organelles, thereby emitting blue and red fluorescence. Moreover, it is reported to allow to read out sub-mitochondrial viscosity.

Indeed, such a probe is missing and I was excited to review this manuscript. The manuscript is very well written.

We thank reviewer for noting the novelty of our work.

Comment 1. *Almost all conclusions in this manuscript rely on the analysis of co-localization. However, for conditions only a single image is shown. No quantifications are reported. For example, already visual inspection of the first image (Fig. 2a, presumably a well-chosen example image), which should report on the co-localization of Coupa-Mito and MTG, is in the referee's eyes not convincingly showing a high level of co-localization of the two signals. Without a solid statistical analysis of the level of co-localization throughout the manuscript, it is impossible to follow the conclusions. In fact, the only statistical analysis is reported in Fig. 3g. Apparently, three manually picked fluorescent dots within a single cell were chosen for statistical analysis. This is just not up to the standards of the field.*

Response: We apologize for the lack of quantification data in previous manuscript. We have made a quantitative analysis of the images involved in the current manuscript, and add these results as new Figure S6, Figure S8, Figure S11 and Figure S16 in Supporting Materials. It reads now:

Line131-135: “To verify the mitochondrial targeting ability of Coupa²⁶, we co-stained the cells with mitochondria-tracker green (MTG) for 30 min, and the followed SIM imaging revealed that blue fluorescent particles were localized in the MTG-labeled mitochondria, **whereas red particles showed a much less co-localization with mitochondria** (Figs. 2a and 2b, dotted white line to show the mitochondrial structure, Fig. S6).” This quantification data further supported our conclusion.

Figure S6 in the revised Supporting Materials.

Line 146-148: “Moreover, the blue fluorescence in the damaged mitochondria was significantly higher than that in untreated HeLa cells (Figs. 2a-d and S8), and this blue fluorescence enhancement appeared mainly on mitochondria membrane” This quantification data proved our statement.

Figure S8 in the revised Supporting Materials.

Line 158-160: “To verify that Coupa-stained red puncta (Fig. 2a,b) labeled lysosomes (Fig. 1b), we

co-stained cells with Coupa and lysosome-tracker green (LTG) in CCCP-treated (Fig. 2e) and untreated cells (Fig. S11).” This quantification data further supported our conclusion.

Figure S11b in the revised Supporting Materials.

Line 210-211: “We then measured the fluorescence intensity distribution of Coupa-mito to reveal the viscosity profile inside mitochondria, and the fluorescence intensity of Coupa-mito staining mitochondria in CCCP-treated cells was accordingly much higher than that of MTG-stained mitochondria (Fig. 4a, S17) confirming the local viscosity enhancement in damaged mitochondria induced by CCCP.” This quantification data supported our statement.

Figure S17 in the revised Supporting Materials.

Comment 2. The authors state that the viscosity of mitochondria increases when they are damaged. This would certainly be interesting. Unfortunately, the paper that is cited for this statement (Lit 16) does not provide any experimental data to support this notion.

Response: We agree with the reviewer that this is an important issue. The theoretical basis to state that the viscosity of mitochondria increases when damaged has also been commented by Reviewer #1 (Reviewer #1’s Comment 3). To further support this notion, we performed an assay to prove this event also occur in our system (CCCP-treated HeLa cells) using a previously reported mitochondria viscosity probe (Peng et al., 2019) (named Mito-V, gifted by Wei-ying Lin group, Jinan university). Compared to untreated HeLa cells (Figure S7a and S7b), the fluorescence intensity of Mito-V-labeled mitochondria (red color) increased (Figure S7c and S7d) in CCCP-treated HeLa cells. This result revealed that the increase in mitochondrial viscosity occurs in mitophagy (Figure S7e and S7f), which is consistent with Coupa-mito’s result. We added this result as the new Figure S7 (line 138-141) in Supporting Materials. It reads now:

“We then sought to determine the capacity of Coupa to differentiate damaged and normal mitochondria based on local viscosity changes via Coupa-405 channel imaging, since it was reported that damaged mitochondria display normally the enhanced local viscosity²⁷, and we confirmed this effect in HeLa cells using a previously reported mitochondria viscosity probe²⁰ (Fig. S7).”

Figure S7 in the revised Supporting Materials.

Moreover, to demonstrate that mitochondria viscosity increases upon damage, we also performed fluorescence lifetime imaging microscopy (FLIM) measurements of Coupa-mito in live cells, which is the gold standard for viscosity probes. As you can see from the new Figure S16, the lifetime of Coupa-mito increased upon CCCP stimulation, indicating that mitochondria increases after damage. This result also confirmed the ability of Coupa to report mitochondria viscosity change. We added this result as the new Figure S16.

Figure S16 in the revised Supporting materials.

Comment 3. The authors report the finding that different concentrations of glycerol (0-80 %) affect the fluorescence intensity of Coupa. They further found that in apoptotic mitochondria Coupa seems to increase in brightness, leading to the conclusion that the viscosity in apoptotic mitochondria changes and Coupa is able to report on these changes. The authors entirely ignore other, much more obvious explanations: The increase could just be a geometrical effect, as during apoptosis thin mitochondrial tubules are transformed into large spheres, which are inherently brighter.

Response: We thank the reviewer for this frank and critical assessment. First of all, since the Coupa-mito is mainly on the mitochondria membrane, the shape change wouldn't be able to induce significant change in the density of fluorophore (Coupa). Secondly, we used super-resolution imaging to report localized intensity increase on some parts of individual mitochondria (Figure 4b and 4g), which cannot simply be caused by geometric change from tubules to spheres. Thirdly, as shown in Figure 3d (middle panel in the right 10 min to 30 min), the shape of mitochondria is approximately a spherical shape to start with but the fluorescent intensity still increased significantly. In addition, our additional FLIM results (line 229-232 in the revised version and Figure S16 in the revised Supporting Materials) confirmed also the enhanced local viscosity in damaged mitochondria. Taken together, these results imply that the change of mitochondrial shape is

unlikely to be the reason for intensity change.

Comment 4. Moreover, it is well documented that binding of MTG to the mitochondrial inner membrane is depending on the membrane potential, which breaks down upon apoptosis. Hence measuring the ratio between Coupa and MTG appears not to be the best measure.

Response: Please refer to Review1's Comment5, we also tested a membrane potential independent mitochondria-GFP to label mitochondria. Next, we used mitochondria-GFP as a control to examine ratio between Coupa-mito and mitochondria-GFP with or without CCCP treatment. The fluorescent intensity of Coupa-mito also increased after CCCP-treatment. We added this result as the new Figure S10 (line 149-154) in Supporting Materials. It reads now:

“To exclude the effect of mitochondria membrane potential (MMP), we co-stained the mitochondria with Coupa and a MMP-independent mitochondria probe, mitochondria-GFP²⁵, and imaging with or without CCCP-altering MMP demonstrated that Coupa labeling mitochondria is not dependent on MMP, similar to that of mitochondria-GFP (Fig. S9), and CCCP-treatment made the blue fluorescence increase with that of the MMP-independent probe GFP (Fig. S10).”

Figure S10 in the revised Supporting Materials.

Comment 5. It is surprising that alternative and obvious other explanations for the change of the fluorescence intensity of Coupa are not addressed. This makes the further use of Coupa to analyze sub-mitochondrial viscosity even more questionable.

Response: The theoretical basis for the viscosity of mitochondria increases in mitophagy, please refer to Reviewer#1's Comment3. Coupa's rotatable C_{sp2}-C_{sp2} bond can respond to medium viscosity showing fluorescence change in different environments. Moreover, we also performed fluorescence lifetime determination experiments in solution, which is the gold standard for viscosity probe. As shown in new Figure 1e and 1h, we found that the lifetime was increased with the increase of viscosity, indicating that our probe is able to report viscosity. We add this result as the new Figure 1e and 1h. It reads now (line 117-123):

“Higher viscosity caused enhanced fluorescence intensity under both 405-nm (Fig. 1d) and 560-nm excitation (Fig. 1g), likely arising from the limited C_{sp2}-C_{sp2} rotation induced by the enhanced medium viscosity. This result suggests that both emission peaks of Coupa were sensitive to viscosity. This mechanism was confirmed by fluorescence lifetime determination (Fig. 1e and 1h), which revealed that both fluorescence lifetimes for coumarin and hemicyanine emissions increased with the increase of viscosity. Moreover, these results demonstrated also that the Coupa's fluorescence intensity correlates with the local medium viscosity.”

Figures 1e and 1h in the revised manuscript.

Comment 6. Indeed, it is unclear why the EM image (Fig. 3i) shows a crista-to-crista distance of about 100 nm, whereas in the SIM-image (Fig. 4b) the crista-to-crista distance seem to be rather at 1000 nm. Also here, for valid conclusions, a sound statistical analysis would be essential.

Response: Thank you very much for your comments, we have added a mitochondrial crista-to-crista distance data analysis ($n = 60$) based on the images captured under SIM, and also applied the reported algorithm (Shao et al., 2020) to accurately calculate this value (150 ± 44 nm), which is similar to the previously reported results under STED microscopy (Stephan et al., 2019). We added this result as the new Figure S18 in Supporting Materials. It reads now (line 238-242):

“Colocalization imaging via SIM revealed that the entire mitochondrial cristae structure could be clearly displayed by MTG fluorescence with the cristae-to-cristae distance ranging from 100 to 200 nm (Figs. 4b and S18), whereas only a few areas of the mitochondrial cristae were stained by Coupa-mito, which confirmed that Coupa targets the inner membrane of mitochondria (Figs. 1b and 4b)”.

Figure S18 in the revised Supporting Materials.

Reviewer #3 (Remarks to the Author):

The manuscript by Chen et al describes a new fluorescent probe that targets mitochondria and lysosomes in different emission colors. The probe is based on a hemicyanine derivative, which exhibits dual emission dependent on the environment. The dual intracellular localization is claimed to help visualizing interaction of mitochondria with lysosomes during mitophagy. Although the results look interesting, the data interpretation is speculative. Generally, the authors do not provide sufficient experimental evidence for their claims. Therefore, I do not recommend this manuscript for publication in Nature Communications. My detailed arguments are given below. After some revisions it could be recommended to another journal, such as Scientific Reports.

Response: We thank the reviewer's comment on the importance of our works. To solidify our conclusion, additional experiments to support viscosity sensing such as fluorescence lifetime determination in solution (new Figure 1e and 1h) and FLIM in live cells (new Figure S16), which is the gold standard for determining viscosity probe have been carried out. Our results demonstrated that Coupa-mito is a viscosity probe for mitochondria. Meanwhile, we also want to mention that our manuscript not only reported a new dye, but also provided a new strategy for future dye development.

Comment 1. *The key claim of this work is that the new probe is sensitive to viscosity. Indeed, experiments in solvents suggest some sensitivity to viscosity in solvents. However, this sensitivity is limited and it does not really mean that the variation of intensity observed inside the cells are related to viscosity. In fact, the changes in the intensity could be related to multiple reasons, including inhomogeneity in dye local concentration, dye self-quenching, dye aggregation, etc. Therefore, interpretation of data in multiple places of the manuscript (abstract, then in pages 6, 7, 9-12) is speculative. One way to provide the evidence that the observed intensity changes are linked to the viscosity is the fluorescence lifetime imaging (FLIM) combined with calibration measurements of lifetimes in organic solvents. Alternatively, the author can remove/revise claims related to viscosity sensing and submit this manuscript to a different journal.*

Response: We fully agree with the reviewer and added the required information in the revised version of our manuscript. According to the Reviewer's suggestion, we performed FLIM assay to measure the probe's lifetimes in organic solvents (glycerol and methanol). As shown in new Figure 1e and 1h, the fluorescent lifetime spectra of Coupa (10 μ M) in solutions with different viscosity in the organic solvents system, we found that the lifetime was increased with the increase of viscosity. This result clearly indicates that our probe is for reporting viscosity. We add this result as the new Figure 1e and 1h. It reads now (line 117-123):

“Higher viscosity caused enhanced fluorescence intensity under both 405-nm (Fig. 1d) and 560-nm excitation (Fig. 1g), likely arising from the limited C_{sp2}-C_{sp2} rotation induced by the enhanced medium viscosity. This result suggests that both emission peaks of Coupa were sensitive to viscosity. **This mechanism was confirmed by fluorescence lifetime determination (Fig. 1e and 1h), which revealed that both fluorescence lifetimes for coumarin and hemicyanine emissions increased with the increase of viscosity. Moreover, these results demonstrated also that the Coupa's fluorescence intensity correlates with the local medium viscosity.**”

Figures 1e and 1h in the revised manuscript.

Moreover, since we claimed that Coupa-mito is a viscosity probe in live cells, we also performed FLIM experiments in live cells. As you can see from the following new Figure S16, the lifetime of Coupa-mito did increase upon CCCP stimulation, indicating the ability to report viscosity change of mitochondria. We added this result as the new Figure S16. It reads now (line 228-232):

“The viscosity-sensitive feature of Coupa-mito fluorescence (Figs. 1d-e, Fig. S5) prompted us to further investigate Coupa-mito’s ability to monitor the viscosity in individual mitochondria. To avoid the influence of probe concentration, we also performed FLIM measurements in HeLa cells to confirm that Coupa-mito could report viscosity in live cells (Fig. S16), and CCCP treatment significantly enhanced Coupa-mito’s fluorescence lifetime corresponding to the increase of mitochondrial viscosity.”

Figure S16 in the revised Supporting Materials.

Comment 2. Another problem concerns reactive sulfur sp (RSS). The authors showed that the long-wavelength band of the dye can be quenched by RSS, but this quenching is not so efficient. In the figure 1c and 1d, it can be seen that in the presence of RSS, the red emission is still stronger than the blue emission. Therefore, it is hard to understand that this partial quenching can be the main reason for the blue emission in mitochondria (as claimed in pages 5 and 7) without any contribution of the red. It seems that the reason for the blue emission of this dye in mitochondria remains unclear, which poses serious problem for

interpretation of the rest of the results in the present work. This is the second key reason why this work cannot be recommended for the present journal.

Response: We are sorry for the misleading Figs. 1c and 1d, which put the emission spectra excited by different excitations (405 nm and 561 nm) in one diagram. Upon excitation at 405 nm, Coupa's blue emission is always stronger than the red emission (see new Fig. 1c), and the presence of RSSs will make the red emission even weaker. On the other hand, we still saw minor red emission in the presence of only one RSS. However, mitochondria are enriched with multiple RSS species. As we can see from the new Figure 1c, the red emission is almost vanished if both Na_2SO_3 and Na_2S were added.

Similarly, the red emission of Coupa upon 560 nm excitations showed negligible fluorescence in the presence of both Na_2SO_3 and Na_2S . Therefore, the red emission can be effectively quenched by the RSSs enriched in mitochondria.

Figures 1c and 1f in the revised manuscript.

Comment 3. Presentation of electron microscopy (EM) data is misleading. Indeed, Figure 3i shows mitochondria and lysosomes, but the authors cannot claim that the structures observed in EM “correlate” with emission of the probe. The authors have not performed correlative EM-fluorescence measurements, therefore mentioning this correlation is the overstatement.

Response: We agree with the reviewer’s suggestion. We have removed the EM results.

Comment 4. In all figure captions presenting multi-color images, the details on the excitation and emission wavelength should be provided.

Response: We have added the detailed information for excitation and emission wavelength in all Figure captions.

Figure 2. MTG channel: Ex, 488 nm, Em, 500-550 nm; LTG channel: Ex, 488 nm, Em, 500-550 nm; Coupa-lyso channel: Ex, 561 nm, Em, 570-640 nm, and Coupa-mito channel: Ex, 405 nm, Em, 420-495 nm.

Figure 3. DAPG channel: Ex, 488 nm, Em, 500-550 nm; Coupa-lyso channel: Ex, 561 nm, Em, 570 -640 nm; Coupa-mito channel: Ex, 405 nm, Em, 420-495 nm.

Figure 4. MTG channel: Ex, 488 nm, Em, 500-550 nm; Coupa-lyso channel: Ex, 561 nm, Em, 570-640 nm; and Coupa-mito channel: Ex, 405 nm, Em, 420-495 nm.

Supplementary Figure 7. Mito-V imaging conditions: Ex, 561 nm, Em, 570-640 nm; and MTG imaging conditions : Ex, 488 nm, Em, 500 - 550 nm.

Supplementary Figure 9. Imaging conditions for Mitochondria-GFP: Ex, 488 nm, and Em, 500-550 nm; Coupa-mito channel: Ex, 405 nm and Em, 420-495 nm.

Supplementary Figure 10. Mitochondria-GFP channel: Ex, 488 nm, Em, 500 - 550 nm; Coupa-mito channel: Ex, 405 nm, Em, 420-495 nm.

Supplementary Figure 11. LTG channel: Ex, 488 nm, Em, 500-550 nm; Coupa-lyso channel: Ex, 561 nm, Em = 570-640 nm.

Supplementary Figure 13. Coupa-lyso channel: Ex, 561 nm, Em, 570-640 nm.

Supplementary Figure 14. DAPG channel: Ex, 488 nm, Em, 500 - 550 nm), Coupa-lyso channel: Ex, 561 nm, Em, 570-640 nm.

Supplementary Figure 15. LTR channel: Ex, 561 nm, Em, 570 - 640 nm; DALG channel: Ex, 488 nm, Em, 500-550 nm.

Supplementary Figure 16. Coupa-mito channel: Ex. 405 nm, Em, 420-495 nm.

Supplementary Figure 18. MTG (Ex = 488 nm, Em = 500-550 nm).

Supplementary Figure 19. Coupa-mito channel: Ex, 405 nm, Em, 420-495 nm; MTG channel: Ex, 488 nm, Em, 500-550 nm.

Comment 5. Page 5, Introduction: “The viscosity of mitochondria increases when they are damaged.” This statement requires references to corresponding literature.

Response: Please see our reply to Reviewer#1’s Comment 3. We also confirmed this experimentally with a previously reported mitochondrial viscosity probe(Mito-V) (Peng et al., 2019) and new FLIM experiments in cells (new Figure S16), please refer to Reviewer2’s Comment 2.

Comment 6. Page 5, Introduction: “Although conventional dyes (e.g., mitochondria-tracker and lysosome-tracker probes) report the morphology of mitochondria and lysosomes³, they are insensitive to viscosity and therefore do not track the dynamic changes of an organelle’s internal environment”. Viscosity sensitive probes for organelles have already been developed (see for instance: Chambers et al ACS Nano 2018, 12, 5, 4398). Therefore, this sentence should be revised and relevant literature should be cited.

Response: We have revised this statement and cited viscosity sensitive probes’ paper (reference list 22,23,24,25, line 110-114). It reads now:

“Although conventional dyes (e.g., mitochondria-tracker and lysosome-tracker probes) report the morphology of mitochondria and lysosomes³, they are insensitive to viscosity and therefore do not track the dynamic changes of an organelle’s internal environment. Recently developed viscosity probes can realize the imaging of the viscosity of a single organelle by using fluorescent probe²² with rotatable C_{sp2}-C_{sp2} bond^{23,24,25}.”

Comment 7. Page 5: “mechanism of multicolour labelling of mitochondria and lysosomes is shown in Fig. 1b”. The authors have not really studied the mechanism of the labelling. Here, they only show a proposed or expected mechanism.

Response: We very much appreciate the reviewer’s suggestion. We also agree with the reviewer that this is an important issue. We have performed assay and proved that the mechanism of Coupa-mito labeling of mitochondria is independent of MMP (please refer to Reviewer 1’s Comment5).

We also performed a series of experiments to verify the mechanism of Coupa-lyso labeling lysosomes. We studied the cellular uptake of Coupa-lyso at different temperatures and at different metabolic or endocytosis levels. As shown in the following new Figure S13, after incubation at 37°C for 30 min, cells showed obvious fluorescence, while at 4°C the fluorescence is significantly lower than that at 37°C. In addition, the use of the metabolic inhibitors (MI, i.e., 2-deoxy-D-glucose and oligomycin) or endocytosis

inhibitor NH_4Cl pre-treatment also reduced intracellular fluorescence intensity (Figure S13). This result implies that the probe entered the cells via energy dependent endocytosis and, as such, passively targeted lysosomes. We added this result as the new Figure S13 in Supporting Materials. It reads now (line 167-171):

“Besides, the lysosome-labeling by Coupa was clarified by Coupa-561 channel imaging at different temperatures, metabolic, and endocytosis levels, implying that the probe entered the cells via energy-dependent endocytosis and, as such, passively targeted lysosomes (Fig. S13). These results suggest that Coupa-labeled red particles can reflect lysosome biogenesis in physiological and pathological conditions.”

Figure S13 in the revised Supporting Materials.

References

- Chen, Y., Zhu, C., Cen, J., Bai, Y., He, W., and Guo, Z. (2015). Ratiometric detection of pH fluctuation in mitochondria with a new fluorescein/cyanine hybrid sensor. *Chemical science* 6, 3187-3194.
- Li, X., Li, X., and Ma, H. (2020). A near-infrared fluorescent probe reveals decreased mitochondrial polarity during mitophagy. *Chemical Science* 11, 1617-1622.
- Miyake, K., Bekisz, J., Zhao, T., Clark, C.R., and Zoon, K.C. (2012). Apoptosis-inducing factor (AIF) is targeted in IFN- α 2a-induced Bid mediated apoptosis through Bak activation in ovarian cancer cells. *Biochimica et Biophysica Acta* 1823, 1378-1388.
- Peng, M., Yin, J., and Lin, W. (2019). Tracking mitochondrial viscosity in living systems based on a two-photon and near red probe. *New Journal of Chemistry* 43, 16945-16949.
- Ren, M., Deng, B., Zhou, K., Kong, X., Wang, J., and Lin, W. (2017). Single Fluorescent Probe for Dual-Imaging Viscosity and H₂O₂ in Mitochondria with Different Fluorescence Signals in Living Cells. *Analytical Chemistry* 89, 552-555.
- Sabharwal, S.S., and Schumacker, P.T. (2014). Mitochondrial ROS in cancer: initiators, amplifiers or an Achilles' heel? *Nature Reviews Cancer* 14, 709-721.
- Shao, X., Chen, Q., Hu, L., Tian, Z., Liu, L., Liu, F., Wang, F., Ling, P., Mao, Z.-W., and Diao, J. (2020). Super-resolution quantification of nanoscale damage to mitochondria in live cells. *Nano Research*.
- Sommer, A.P., Haddad, M., and Fecht, H.J. (2015). Light Effect on Water Viscosity: Implication for ATP Biosynthesis. *Scientific reports* 5, 12029-12029.
- Stephan, T., Roesch, A., Riedel, D., and Jakobs, S. (2019). Live-cell STED nanoscopy of mitochondrial cristae. *Scientific reports* 9, 12419.
- Sun, W., Shi, Y., Ding, A., Tan, Z., Chen, H., Liu, R., Wang, R., and Lu, Z. (2018). Imaging viscosity and peroxynitrite by a mitochondria-targeting two-photon ratiometric fluorescent probe. *Sensors and Actuators B-chemical* 276, 238-246.
- Yang, Z., Cao, J., He, Y., Yang, J.H., Kim, T., Peng, X., and Kim, J.S. (2014). Macro-/micro-environment-sensitive chemosensing and biological imaging. *Chemical Society Reviews* 43, 4563-4601.

REVIEWER COMMENTS

Reviewer #1 (Remarks to the Author):

The author has made necessary supplement and modification, thus it is suggested to accept it.

Reviewer #2 (Remarks to the Author):

The authors have made several amendments to the previous version of the manuscript. I am still not convinced that the key statements are sufficiently substantiated by the data.

My main concern was and is the statistical analysis of the imaging data. In the revised version of the manuscript, quantifications are provided. However, no information, neither in the main text nor in the materials and methods section is given on how this quantification was performed. No indications on what kind of statistical tools were used. As errors, the standard error of the mean (SEM) was chosen in most cases. No indication is given why this was preferred over the standard deviation. According to the figure legends, for statistical analysis just 3 to 5 images were used in most cases. This is by far not sufficient, especially as it is straightforward to record hundreds of such images. Why did the authors use so few images for analysis? Are these pre-selected images? As the statistical analysis is key to the paper, I do not support publication of the manuscript.

Reviewer #3 (Remarks to the Author):

In the revised manuscript, the authors addressed well all my concerns and made sufficient number of new experiments to support their claims. Now I can recommend this manuscript for publication in Nature Communications in the present form.

Response to the review comments

We thank the Reviewer #1 and #3 for accepting our manuscript for publication. We also thank the Reviewer #2 for his/her constructive criticism. We have revised the manuscript accordingly and all of his/her concerns were addressed by appropriate revision of the text. We believe the manuscript greatly improved in its rigor and presentation thanks to the reviewers' help.

Reviewers' comments:

Reviewer #1

The author has made necessary supplement and modification, thus it is suggested to accept it.

We thank the reviewer for the positive and encouraging notes on our efforts to improve the manuscript.

Reviewer #2

The authors have made several amendments to the previous version of the manuscript. I am still not convinced that the key statements are sufficiently substantiated by the data.

We thank the reviewer for the positive note on our manuscript, and appreciate valuable comments.

Comment 1. My main concern was and is the statistical analysis of the imaging data. In the revised version of the manuscript, quantifications are provided. However, no information, neither in the main text nor in the materials and methods section is given on how this quantification was performed. No indications on what kind of statistical tools were used.

Response: With help from our Statistics Consulting Center and Dr. Yanan Zhao, a statistical expert at Shandong University, we included more data to improve our analysis. Detailed “data analysis” has been added in the “Methods” section.

“Data analysis

Statistical analysis was performed with Prism 8 (GraphPad). Normality and lognormality test was conducted. In the case of normal distribution, the statistical comparison of results was checked with a Student’s *t* test. In the case of non-normal distribution, the statistical comparison of results was checked with a Mann-Whitney test. The levels of significance were set at n.s. (no significant difference), * $P < 0.05$, ** $P < 0.01$, *** $P < 0.001$, and **** $P < 0.0001$. Data are presented as mean \pm SEM. SEM was used to compare experimental results with controls³³. For co-localization analysis and average fluorescence intensity, we used individual fluorescent spots or randomly divided cells into four sub-cellular regions for quantitative analysis. Analyzed cells were obtained from three replicates. Statistical significances and sample sizes in all graphs are indicated in the corresponding figure legends. All associated data points can be found in the Source Data file.”

Comment 2. As errors, the standard error of the mean (SEM) was chosen in most cases. No indication is given why this was preferred over the standard deviation.

Response: As written in a guidance titled “Error bars in experimental biology” published in *JCB* (Ref ¹), “Rule 4: because experimental biologists are usually trying to compare experimental results with controls, it is usually appropriate to show inferential error bars, such as SE or CI, rather than SD.” We added the following sentence in the “Data analysis”.

“Data are presented as mean \pm SEM. SEM was used to compare experimental results with controls³³.”

Comment 3. According to the figure legends, for statistical analysis just 3 to 5 images were used in most cases. This is by far not sufficient, especially as it is straightforward to record hundreds of such images. Why did the authors use so few images for analysis? Are these pre-selected images? As the statistical analysis is key to the paper, I do not support publication of the manuscript.

Response: We didn’t select image for analysis. 3-5 images containing several cells were obtained from the same measurement. In recent similar studies published in *Nature*² and *Nature Communications*^{3,4} using SIM super-resolution microscopy for imaging sub-cellular dynamics of individual cells, the authors used the sub-cellular regions in 2017 *Nature Communications* (Ref ⁴), contacts in 2018 *Nature* (Ref ²) or events in 2020 *Nature Communications* (Ref ³) for quantitative analysis. We adapted a similar approach. Four random regions from each cell or individual fluorescent spots were used for co-localization and average fluorescence intensity analysis. We re-analyzed data throughout the manuscript (new Fig. 3g, Fig. S6, Fig. S7f, Fig. S8, Fig. S9c, Fig. S10c, Fig. S11b, Fig. S12, Fig. S13b, Fig. S15f, Fig. S17). 10-29 cells for each group were used in the revision. Our sample size is comparable with what have been used in similar studies published in *Nature* (9-26 cells in Ref ²) and *Nature Communications* (10 cells in Ref ³ and 10-20 cells in Ref ⁴). Moreover, we also included every data points in the plots as well as the source data file associated with the manuscript. We believe the new statistical analysis in our revision is rigorous enough to support our conclusions.

g
Figure 3. Coupa monitors mitochondrion–lysosome interactions in mitophagy via SIM imaging. (a) Co-localization of Coupa-stained mitochondria/lysosomes and DAPG-stained autophagosomes in CCCP-treated (10 μ M, 12 h at 37 $^{\circ}$ C) HeLa cells. (b) Zoom-in images of regions of interest in white frames representing the mitochondria inside (1) and outside (2) the autophagosomes. (c) Zoom-in images of regions of interest in white frames representing the lysosomes inside (3) and outside (4) the autophagosomes. (d) SIM tracking of mitochondrion-lysosome interaction during mitophagy in the CCCP-treated (50 μ M) HeLa cells via Coupa staining; solid white lines in the merged images indicate where fluorescence intensity profiles shown in (e) were measured. (f) SIM imaging of mitochondrion–lysosome interaction in CCCP-treated cells for 12 h; white rectangles indicate lysosomes outside autolysosomes (1), mitochondria and lysosomes inside autolysosomes (2), and mitochondria outside autolysosomes (3). (g) Normalized average fluorescence intensity of mitochondrion–lysosome interactions. Data are mean \pm SEM ($n = 44-49$ particles from 10-15 cells, **** $P < 0.0001$). (h) Schematic representation of Coupa for monitoring mitochondria–lysosome interaction in mitophagy. DAPG channel: Ex, 488 nm, Em, 500-550 nm; Coupa-lyso channel: Ex, 561 nm, Em, 570 -640 nm; Coupa-mito channel: Ex, 405 nm, Em, 420-495 nm.

Supplementary Figure 6. Co-localization coefficients of MTG-labeled mitochondria with the Coupa-labeled blue and red fluorescent particles. Data are mean \pm SEM ($n = 64-68$ areas from 16-17 cells, **** $P < 0.0001$).

e
Supplementary Figure 7. SIM imaging of Mito-V-labeled mitochondria in HeLa cells with or without CCCP treatment. (a) Mitochondria co-stained with MTG and Mito-V in untreated (a) and CCCP-treated HeLa cells (c). (b and d) Zoom-in images of white rectangles in (a) and (c). (e) Normalized mean intensity of Mito-V stained mitochondria with or without CCCP treatment. Data are mean \pm SEM ($n = 48-64$ areas from 12-16 cells, **** $P < 0.0001$). (f) Schematic representation of Mito-V-labeled mitochondria with or without CCCP treatment. Mito-V imaging conditions: Ex, 561 nm, Em, 570-640 nm; and MTG imaging conditions: Ex, 488 nm, Em, 500 - 550 nm.

Supplementary Figure 8. Normalized mean fluorescence intensity of Coupa-labeled blue fluorescent particles in untreated and CCCP-treated HeLa cells. Data are mean \pm SEM ($n = 80$ areas from 20 cells, **** $P < 0.0001$).

Supplementary Figure 9. SIM imaging of CCCP-treated HeLa cells co-stained by Coupa and GFP via a coupa-mito/GFP dual channel mode. (a) Schematic representation of experiments. (b) Co-localization imaging of GFP and Coupa (Coupa-mito channel) before or after CCCP treatment, white rectangles indicate Zoom-in images of areas. (c) Co-localization coefficients of Coupa-mito with GFP. Data are mean \pm SEM ($n = 40$ areas from 10 cells for each group, n.s. for no significant difference). Imaging conditions for Mitochondria-GFP: Ex, 488 nm, and Em, 500-550 nm; Coupa-mito channel: Ex, 405 nm and Em, 420-495 nm.

Supplementary Figure 10. SIM imaging of HeLa cells co-stained by Coupa and mitochondria-GFP with or without CCCP treatment. (a and b) Mitochondria co-stained with Coupa-mito and Mitochondria-GFP in untreated (a) and CCCP-treated HeLa cells (b), white rectangles indicate Zoom-in images of region of interests. (c) Quantitative analysis of the average fluorescence intensity of Coupa-mito and mitochondria-GFP with or without CCCP treatment. Data are mean \pm SEM ($n = 52-88$ areas from 13-22 cells for each group, n.s. for no significant difference and **** $P < 0.0001$). Mitochondria-GFP channel: Ex, 488 nm, Em, 500 - 550 nm; Coupa-mito channel: Ex, 405 nm, Em, 420-495 nm.

Supplementary Figure 11. The overlap images of Coupa-labeled red particles and lysosome-tracker-green (LTG) labeled lysosomes in untreated HeLa cells (a) and co-localization (b) with structured illumination microscopy (SIM). Data are mean \pm SEM ($n = 116$ areas from 29 cells). LTG channel: Ex, 488 nm, Em, 500-550 nm; Coupa-lyso channel: Ex, 561 nm, Em = 570-640 nm.

Supplementary Figure 12. Characterization of Coupa-labeled red fluorescent particles in live HeLa cells. (a) The diameters of Coupa-labeled red fluorescent particles in HeLa cell. Data are mean \pm SEM ($n = 158$ particles from 10 cells). (b) The count of Coupa-labeled red fluorescent particles. Data are mean \pm SEM ($n = 20-35$ cells, **** $P < 0.0001$).

Supplementary Figure 13. Investigation of the cell uptake of Coupa by HeLa cells via SIM imaging. (a) SIM images of Coupa-lyso labeled lysosomes at 37°C, 4°C, and with MI or NH₄Cl treatment, and quantitative analysis was shown in (b). Data are mean \pm SEM ($n = 44-64$ areas from 11-16 cells, **** $P < 0.0001$). Coupa-lyso channel: Ex, 561 nm, Em, 570-640 nm.

f

Supplementary Figure 15. Co-localization between lysosomes and autolysosomes in CCCP-treated HeLa cells stained by Coupa and LTR via SIM imaging. (a) Schematic representation and SIM imaging of co-localization of lysosomes stained with commercial lysosome tracker red (LTR) and DALG-stained autolysosomes in CCCP-treated HeLa cells. (b) Representative fusion of Coupa-lyso-stained lysosomes and DALG-stained autolysosomes; and (c) schematic representation of commercial LTR unsuitable to distinguish lysosomes from autolysosomes. (d) Co-localization of Coupa-lyso-stained lysosomes and DALG-stained autolysosomes in CCCP-treated HeLa cells; white rectangle represents amplification shown in (e). (f) Co-localization coefficients of Coupa-lyso with LTR labeled lysosomes or DALG labeled autolysosomes. Data are mean \pm SEM ($n = 40-52$ areas from 10-13 cells, **** $P < 0.0001$). LTR channel: Ex, 561 nm, Em, 570 - 640 nm; DALG channel: Ex, 488 nm, Em, 500-550 nm

Supplementary Figure 17. The normalized mean intensity of Coupa-mito-stained and MTG-stained mitochondria in CCCP-treated HeLa cells. Data are mean \pm SEM ($n = 80$ areas from 20 cells for each group, **** $P < 0.0001$).

Reviewer #3

In the revised manuscript, the authors addressed well all my concerns and made sufficient number of new experiments to support their claims. Now I can recommend this manuscript for publication in Nature Communications in the present form.

We thank the reviewer for the valuable comments on our manuscript.

Reference

1. Cumming GD, Fidler F, Vaux DL. Error bars in experimental biology. *Journal of Cell Biology* **177**, 7-11 (2007).
2. Wong YC, Ysselstein D, Krainc D. Mitochondria–lysosome contacts regulate mitochondrial fission via RAB7 GTP hydrolysis. *Nature* **554**, 382-386 (2018).
3. Qin J, *et al.* ER-mitochondria contacts promote mtDNA nucleoids active transportation via mitochondrial dynamic tubulation. *Nature Communications* **11**, 4471 (2020).
4. Han Y, Li M, Qiu F, Zhang M, Zhang Y-H. Cell-permeable organic fluorescent probes for live-cell long-term super-resolution imaging reveal lysosome-mitochondrion interactions. *Nature communications* **8**, 1307 (2017).

REVIEWERS' COMMENTS

Reviewer #2 (Remarks to the Author):

The authors have addressed my concerns.

Response to the review comments (3rd round)

Reviewer #2 (Remarks to the Author):

The authors have addressed my concerns.

We thank the reviewer for the positive and encouraging note on our efforts to improve the manuscript.